# Flexibility and Adaptation of Cancer Cells in a Heterogenous Metabolic Microenvironment

**DOI:** 10.3390/ijms22031476

**Published:** 2021-02-02

**Authors:** Gabriele Grasmann, Ayusi Mondal, Katharina Leithner

**Affiliations:** 1Division of Pulmonology, Department of Internal Medicine, Medical University of Graz, A-8036 Graz, Austria; gabriele.grasmann@medunigraz.at (G.G.); ayusi.mondal@medunigraz.at (A.M.); 2BioTechMed-Graz, A-8010 Graz, Austria

**Keywords:** cancer, metabolic microenvironment, heterogeneity, adaptability, gluconeogenesis

## Abstract

The metabolic microenvironment, comprising all soluble and insoluble nutrients and co-factors in the extracellular milieu, has a major impact on cancer cell proliferation and survival. A large body of evidence from recent studies suggests that tumor cells show a high degree of metabolic flexibility and adapt to variations in nutrient availability. Insufficient vascular networks and an imbalance of supply and demand shape the metabolic tumor microenvironment, which typically contains a lower concentration of glucose compared to normal tissues. The present review sheds light on the recent literature on adaptive responses in cancer cells to nutrient deprivation. It focuses on the utilization of alternative nutrients in anabolic metabolic pathways in cancer cells, including soluble metabolites and macromolecules and outlines the role of central metabolic enzymes conferring metabolic flexibility, like gluconeogenesis enzymes. Moreover, a conceptual framework for potential therapies targeting metabolically flexible cancer cells is presented.

## 1. Introduction

The extracellular metabolic milieu in tumors, also referred to as the metabolic microenvironment, has a major impact on the metabolic phenotype of cancer cells, as well as on the responsiveness of tumor-invading immune cells [1,2,3]. It is largely determined by the blood supply, however, tumor cells and non-neoplastic cells of the tumor microenvironment also shape the metabolic landscape in the tumor due to consumption of nutrients and release of waste and other metabolites [1,2,3]. These interactions between cancer cell metabolic activity and extratumoral metabolites are reminiscent of the concept of plasticity and reciprocity observed for cancer cell interaction with the extracellular matrix and non-neoplastic cells, like fibroblasts [4]. According to this concept, cancer cell invasion is governed by mechanic and signaling-related cues from the extracellular matrix and from normal cells. The key feature of this process is the adaptability of cancer cells, allowing positional and phenotypic changes and promoting cancer progression [4]. Genomic instability, epigenetic changes and functional plasticity all have been implicated in plasticity, ultimately leading to inter- and intralesion heterogeneity [4]. Plasticity and reciprocity are not specific for cancers, they are central also to wound healing or epithelial morphogenesis, however in cancers, these processes are perpetuated [4]. Cancer cells, similar to normal cells in the body, adapt to the prevailing metabolic conditions in their environment [5,6]. The metabolic microenvironment of solid cancers is heterogenous, comprising areas of hypoxia and nutrient scarcity due to an insufficient vascular network. These conditions may impose severe stress on neoplastic tumor cells, triggering significant adaptations [3,4,7]. Two modes of cancer cell metabolic switching have been defined: flexibility (the flexible use of different fuels) and plasticity (the utilization of certain nutrients via different metabolic pathways) [8], which likely occur in parallel. Especially when cancer cells relocate to form metastases, considerable adaptation and flexibility are required due to the strikingly different microenvironmental composition in distant tissues compared to the parental tissue. The role of metabolic plasticity and flexibility of cancer cells during cancer progression and metastasis has been highlighted by recent excellent reviews on the topic [8,9]. Here, we discuss the recent literature on metabolic flexibility of tumor cells, focusing on the flexible utilization of nutrients and on enzymes conferring metabolic flexibility under low-glucose conditions, especially gluconeogenesis enzymes. In contrast to previous reviews discussing the role of gluconeogenesis and pyruvate metabolism in glucose-deprived cancer cells [10,11,12], the present review sheds light on the context dependent utilization of different alternative fuels via gluconeogenesis-dependent and independent pathways. Moreover, we discuss potential therapeutic strategies to target metabolically flexible cancer cells.

## 2. Blood Perfusion and the Metabolic Microenvironment

The extracellular metabolic milieu of cancer cells is determined by numerous factors, including blood perfusion, nutrient uptake and waste secretion by tumor cells and non-neoplastic cells in the tumor microenvironment, and drainage by veins and lymphatics [1,2]. The heterogeneity of the extracellular milieu within a tumor, comprising soluble factors like nutrients or oxygen and insoluble factors like the extracellular matrix, has an important impact on cancer cell metabolism [3]. Cancer cell metabolism may also be influenced by the neighboring normal tissue, e.g., in the brain, it is characterized by high concentrations of glutamate [13]. Importantly, cancer cells disseminating via the lymphatics or in cavities like the pleural or peritoneal space encounter very specific metabolic conditions present in these anatomic sites [14,15,16]. The metabolic microenvironment is further altered during metastatic dissemination. For example, during peritoneal metastasis, cancer cells come into contact with resident visceral adipocytes and reprogram these to export fatty acids eventually used by the tumor cells [16,17].

Although angiogenesis, the growth of new blood vessels, is induced already in small tumors, the supply with nutrients and oxygen is frequently poor in solid cancers. The newly formed vascular network is aberrant and irregular, resulting in poor perfusion, fluctuating blood flow and increased interstitial pressure [18,19,20,21]. In fact, dynamic contrast material-enhanced (DCE) magnetic resonance imaging shows a very heterogenous pattern of perfusion of solid cancers like breast cancer and lung cancer [22,23]. The spatial and temporal microvascular insufficiency and the rapid consumption by the proliferating cancer cells both result in steep gradients for glucose and other nutrients [18,19,20,21]. Together, these factors lead to a highly heterogenous metabolic tumor microenvironment (Figure 1). Importantly, nutrient availability is one of the most important determining factors for cancer cell metabolism and acts together with intrinsic factors, such as activation of oncogenes or mutations of genes encoding for metabolic enzymes [24,25]. On the other hand, the selection for certain oncogenes may be favored by stress conditions in the extracellular milieu, e.g., glucose limitation [26].

## 3. Availability of Nutrients in Solid Cancers

Although glucose is an important nutrient for cancer cells, the poor vascular supply and high rate of consumption lead to a poor supply of this sugar in certain parts of solid cancers [1,19]. As a paradox, glucose uptake and withdrawal of glucose from the vasculature are enhanced in many cancer types, as may be visualized by [18F] fluorodeoxyglucose positron emission tomography (FDG-PET). As shown in numerous studies, glucose concentration is consistently lower in cancers than in the corresponding normal tissues [27,28,29,30]. This corresponds to a lower average concentration of glucose in the tumor extracellular fluid, as found in a recent study using a murine pancreatic cancer model [31]. In this study, the extracellular fluid was sampled by rapid centrifugation of fresh cancer tissue, carefully controlling for cancer cell lysis. In comparison to plasma, extracellular fluid from pancreatic tumors showed reduced levels of glucose with a mean reduction of approximately 50%, and lower levels of amino acids tryptophan, arginine and cystine, while glutamine levels were not altered [31]. The composition of the metabolic cancer microenvironment and the adaptive responses in cancer cells to these specific local nutritional conditions are currently under intense investigation, since these might represent specific vulnerabilities in cancers that could potentially be exploited for therapy. Moreover, according to present concepts, vascular insufficiency, though putting constraints to unlimited proliferation and growth, may pose an advantage for the tumor in the context of treatment, since tumor cells able to survive in these regions may become difficult to treat due to a lack of drug delivery [9]. Hypoxia, reduced oxygen concentration, is another important consequence of poor perfusion. Its contribution to shaping cancer cell metabolism and favoring the emergence of more aggressive phenotypes is being extensively studied (for review, see [32]). In contrast, the role of cancer cell adaptation to nutrient deprivation and the resulting metabolic flexibility for cancer progression are still poorly understood.

## 4. Glucose Metabolism and Oxidative Phosphorylation in Cancer—Partners in Crime

Glucose is a major nutrient for most cancer types. It is transported into cancer cells via different glucose transporters, mainly by glucose transporter 1 (GLUT1, encoded by *SLC2A1*) and metabolized via glycolysis [24,25]. Glycolysis provides intermediates for the generation of important cellular building blocks [24,25] and eventually generates pyruvate which is mainly reduced to lactate (Figure 2). In the body, excess glucose is stored as glycogen, mostly in specialized organs like the liver or muscle, but also in other peripheral tissues [33]. Storage of glucose in the form of glycogen also occurs in cancer cells to variable degrees and may lead to a “clear-cell” phenotype. Upon glucose starvation, glycogen can be mobilized, releasing glucose-1-phosphate which is converted to glucose-6-phosphate and further shuttled towards glycolysis [34]. 

The full oxidation of glucose-derived pyruvate in mitochondria via conversion to acetyl-CoA is relatively decreased in cancer cells, partly due to the overexpression of kinases inhibiting pyruvate dehydrogenase [24,25]. The preference of (aerobic) glycolysis over full oxidation of glucose has been observed in tumor slices as early as 1924 by Otto H. Warburg (reviewed in [35]). This so-called Warburg effect (aerobic glycolysis) is found in tumor cells and other highly proliferative cells [35]. Damage to mitochondria, however, is not the cause for the Warburg effect, contrary to the assumptions by O.H. Warburg [35]. Mitochondria are still highly active in cancer cells, producing adenosine triphosphate (ATP) and acting as metabolic factories and signaling organelles [35]. In fact, both glycolysis and oxidative phosphorylation (OXPHOS) are utilized in cancer cells to optimize cancer cell metabolism for unlimited growth [36]. The dependency of many tumor cell lines on OXPHOS in low-glucose conditions has been highlighted in an RNAi (RNA interference)-screen for metabolic liabilities [37].

## 5. The TCA Cycle Is a Metabolic Hub with Major Importance for Metabolic Adaptations

Glycolysis and OXPHOS are tightly connected via the tricarboxylic acid cycle (TCA cycle, Krebs cycle). The TCA cycle represents a set of reactions mediating the complete oxidation of acetyl-CoA into two molecules of CO_2_ and the generation of reducing equivalents oxidized in the respiratory chain to produce the proton gradient required for the production of ATP (Figure 2 and Figure 3). Cancer cells may utilize glucose-derived pyruvate but also other precursors to generate acetyl-CoA [38]. In 2008, exogenous lactate has been proposed as an alternative fuel in cancer cells, since it has been shown to be avidly consumed in a glucose-deprived medium by oxygenated cancer cells [39]. Since then, numerous studies highlighted the flexible utilization of lactate as a respiratory fuel feeding into the TCA cycle in cancer cells (reviewed in [40]). The TCA cycle, however, is also a metabolic hub that generates the precursors for multiple biosynthetic reactions, while being replenished by so-called anaplerosis (Figure 2 and Figure 3). As such, the TCA cycle is the starting point for (1) the generation of fatty acids from citrate, (2) the biosynthesis of aspartate and hence nucleotides, (3) the biosynthesis of other nonessential amino acids required, e.g., for protein biosynthesis. It also generates the precursor for gluconeogenesis and thus for a diversity of biosynthetic pathways branching from gluconeogenic/glycolytic intermediates, as will be discussed in Section 8. Glutamine is one of the most important anaplerotic precursors, fueling the TCA cycle via α-ketoglutarate (see Section 6 below). However, the TCA cycle may also be filled from lactate and pyruvate via pyruvate carboxylase (PC) (see Section 7).

## 6. Glutamine as Anabolic Precursor

In the past years, numerous studies identified amino acids to be critical metabolites for cancer growth (reviewed in [24,25]). The selective dependency of acute lymphoblastic leukemia cells on asparagine led to the routine clinical use of recombinant asparaginase to reduce asparagine levels in the blood, which leads to the death of leukemia cells that are auxotrophic for asparagine [24,25]. Interestingly, despite the high consumption of glucose, amino acids, rather than glucose, have been found to account for the majority of cell mass in proliferating mammalian cells, including cancer cells [41]. 

Upon uptake and degradation, amino acids may either yield acetyl-CoA (ketogenic amino acids) or enter the TCA cycle via anaplerosis (glucogenic amino acids) [42]. Amino acids forming acetyl-CoA do not lead to a net contribution to the cell’s biomass, since the two carbons added to the TCA cycle are balanced by the loss of two carbons as CO_2_ in the first round of the TCA cycle (Figure 3). In contrast, anaplerotic precursors that feed into the regenerating pool of TCA cycle intermediates act as important precursors for anabolic reactions [36]. Glutamine, the most abundant amino acid in plasma, is one of the most important anaplerotic precursors. It is taken up by cancer cells at high rates and diverted towards the TCA cycle after its conversion to glutamate and α-ketoglutarate. Moreover, glutamine is an important nitrogen donor [24]. Also, other amino acids or pyruvate may feed into the TCA cycle, and such a preference for non-glutamine precursors has been noted in the physiological milieu of cancer cells in vivo [43].

## 7. Lactate and Pyruvate as Alternative Fuels and Carbon Sources

As outlined before, hypoxia is frequent in solid cancers due to a lack of perfusion and it is one of the drivers of the glycolytic phenotype in many cancers. It has been observed that most solid tumors consist of heterogenous populations of cells, hypoxic cells at the inner core and more oxygenated cells towards the tumor margin. The hypoxic cells are more dependent on glucose consumption and glycolysis for energy production [39]. In glycolytic cancer cells, a high proportion of glucose is converted to lactate even in aerobic conditions [44]. Lactate is transported across the plasma membrane together with a proton by monocarboxylate transporters (MCTs) 1–4 [45]. The expression of MCT1 and MCT4 has been linked to poor prognosis in different types of cancer [46], and MCT inhibitors inhibited tumor growth in different models [46,47]. The direction of lactate transport by MCTs in cancer cells, however, is dependent on the intracellular and extracellular concentrations of protons and lactate [45]. In certain contexts, lactate import, rather than export, has been shown to be an advantage in cancer cells [39], and the uptake of stable isotopically labeled lactate has been confirmed in human lung cancers by stable isotopic tracing [48,49]. The effects of lactate on cancer cell energetics and the role of MCTs in tumor growth and in modulation of antitumor immunity have been addressed in excellent recent reviews [50,51,52]. The imported lactate is oxidized by lactate dehydrogenase to pyruvate, which may be transported to mitochondria and converted to acetyl-CoA (Figure 3). Acetyl-CoA condenses with the TCA cycle metabolite oxaloacetate (OAA) to form citrate. Eventually, citrate is utilized in the TCA cycle for the production of reducing equivalents used by the respiratory chain. Alternatively, citrate may leave the mitochondria and be converted back to acetyl-CoA, for further use in the fatty acid biosynthetic pathway or in acetylation reactions [53]. Importantly, as detailed in Section 8, lactate may not only serve as an energy fuel and source of acetyl-CoA, it may also be converted to glycolytic intermediates via the gluconeogenesis pathway.

Other than lactate, pyruvate may also be utilized as an alternative fuel and biosynthetic precursor in cancer cells. Hyperpolarized ^13^C-pyruvate is rapidly taken up, e.g., by cancer xenografts in animal models, and studies in lung cancers are on the way [54]. MCT1 has been recently found to mediate the import of intravenously injected hyperpolarized ^13^C-pyruvate into cancer cells [55]. However, plasma levels of pyruvate are way below those of lactate (60 µM compared to approximately 1600 µM) [56] and pyruvate concentration is even decreased in cancer extracellular fluid compared to plasma in a mouse model of pancreatic cancer [31]. Acetyl-CoA may be formed not only from pyruvate, amino acids or fatty acids, but also from acetate. Acetate, a short fatty acid, is converted to acetyl-CoA by an enzyme that has been shown to promote cancer growth in different models, acetyl-CoA synthetase 2 (ACSS2) [57,58,59]. Other than promoting the use of acetate derived from circulation, ACSS2 is also necessary for the reuse of acetate released by histone deacetylation, thus maintaining nuclear acetyl-CoA pools for further acetylation reactions [60].

Glutamine and lactate thus represent important alternative fuels, however, their use in cancer cells is not limited to conditions of glucose deprivation. Glutamine, which is also an important nitrogen source, is used by many cancers to fuel the TCA cycle despite the presence of glucose (reviewed in [36]). This contribution might be lower in cancer cells showing high PC flux from pyruvate to OAA, as has been described, e.g., in non-small cell lung cancer [61] and lung metastases of breast cancer [62]. The simultaneous uptake of glucose and glutamine might enhance overall anabolic capacities in highly proliferative cells. In fact, it has been shown that glucose and glutamine uptake influence each other in mammalian cells, albeit in cell type-specific manner [63]. Similarly, lactate has been found to be utilized for the biosynthesis of lipids via pyruvate and acetyl-CoA formation also in the presence of glucose in different cancer cell types [64]. Lactate use and oxidation, on the other hand, contribute not only to carbon and energy balance, but also to maintaining a reduced oxidation state by generating NADH [53]. Thus, the use of glucose and non-glucose carbon sources may occur simultaneously in cancer cells, however, glucose limitation induces greater dependency on the alternative pathways. 

## 8. Gluconeogenesis Enhances Metabolic Flexibility and Anabolism Under Glucose Deprivation

Gluconeogenesis, the formation of glucose from non-carbohydrate precursors like amino acids or lactate, occurs primarily in the liver and kidney [65]. It implies the conversion of precursors like amino acids and lactate towards glycolytic intermediates and further to glucose in a set of reactions that are largely the reverse of glycolysis. As has been shown recently, lactate and glutamine contribution to glycolytic intermediates, in fact, occurs broadly in different tissues of the body, especially in a starved, but also in a fed state [33]. In non-gluconeogenic tissues, the gluconeogenesis pathways only proceed partially, and no glucose is formed form the glycolytic intermediates. Instead, the glycolytic intermediates are utilized by the cells themselves. The steps involved in such partial gluconeogenesis are depicted in Figure 3. It has been previously known that partial gluconeogenesis, contributes to the biosynthesis of glycerol phosphate in hepatocytes and adipocytes, where it promotes the re-esterification of fatty acids to form neutral lipids [66]. Thus, “cell-autonomous” utilization of gluconeogenesis intermediates, as opposed to the classical route of gluconeogenesis leading to the release of glucose, is a physiological process. However, as explained below, this pathway is also exploited by cancer cells to flexibly generate anabolic intermediates for biomass generation from non-carbohydrate precursors.

The first step of gluconeogenesis is mediated by phosphoenolpyruvate carboxykinase (PEPCK), which catalyzes the conversion of OAA to phosphoenolpyruvate (PEP). Two isoforms of PEPCK exist, a cytoplasmic isoform PCK1 (PEPCK-C), and a mitochondrial isoform PCK2 (PEPCK-M) [67,68]. Initially, the existence of (partial) gluconeogenesis in cancer cells not originating from a classical gluconeogenic organ was described in lung cancer cells in 2014 [69]. Lung cancer cells treated with low-glucose, serum-free media showed upregulation of PCK2 and conversion of stable isotopically labelled lactate along the PCK2 pathway to PEP [69]. Silencing or inhibition of the pathway impaired lung cancer cell survival under glucose deprivation and 3D cancer spheroid growth. Subsequently, PCK2 or PCK1 have been found to be active especially under conditions of glucose deprivation to promote cell survival or proliferation in diverse cancer types [69,70,71,72,73,74,75,76,77,78,79,80]. Inhibition of cancer growth by inhibition of PCK1 or PCK2 has been shown in different tumor models in vivo, including xenografts of lung cancer, prostate cancer and colon cancer [71,73,74,77,78,81]. However, in liver cancer and hepatocellular carcinoma, PCK1 or PCK2 have been found to be downregulated and to inhibit cancer cell proliferation or tumor growth in vivo [82,83,84,85]. The view that gluconeogenic enzymes are rather tumor-suppressive in cancers arising in these organs was challenged by a recent study that uncovered a novel function of PCK1 in hepatocellular carcinoma, both in vitro and in vivo. The group showed that upon phosphorylation by AKT, PCK1 translocates to the endoplasmic reticulum, where it mediates the GTP-dependent phosphorylation of the regulatory INSIG proteins, with a resulting hyperactivation of sterol regulatory element-binding proteins (SREBPs) and lipogenesis, thereby stimulating tumor growth [86]. The downstream metabolites generated via PCK1 or PCK2 in glucose-deprived cancer cells include all the biosynthetic precursors classically considered to be derived from glucose: the phospholipid glycerol backbone [77], serine and glycine used for purine biosynthesis [71,87] and ribose phosphate [72,73] (Figure 3). Thus, PCK1 or PCK2 enable cells to produce gluconeogenic/glycolytic intermediates for biosynthetic pathways that are essential for cancer cell proliferation in the absence of glucose. Moreover, PCK2 was shown to directly impact TCA cycle function in cancer cells by catabolizing OAA generated from glutamine [74], although this function was studied in high glucose. PCK2 was shown to enhance the levels of PEP also in the presence of glucose in colon carcinoma cells, with a resulting activation of the nuclear factor of activated T-cells (NFAT) and MYC signaling pathways [80]. The diverse functions of PCK1 and PCK2 in cancer cells, either promoting partial gluconeogenesis, regulating the TCA cycle or in moonlighting, nonclassical functions are not fully elucidated. However, the gluconeogenesis pathway appears to be necessary for the survival and proliferation of glucose-limited tumor cell subpopulations in diverse cancers, thereby contributing to metabolic flexibility. Inhibition of the responsible enzymes, PCK1 or PCK2, thus might interfere with adaptation and suppress tumor growth. Still, more research is warranted to understand the role of PCK1 and PCK2 in different tumor entities.

The endoplasmic reticulum stress mediator cyclic AMP-dependent transcription factor ATF4 plays an important role in the activation of PCK2 expression upon glucose or glutamine deprivation in cancer cells [70]. Moreover, PCK2 expression has been shown to be enhanced by binding of purine-rich element binding protein α (PUR-α) to the PCK2 promoter [88] and by the MYC superfamily member MondoA [89]. Simultaneous silencing of hypoxia-inducible factors (HIFs), HIF-1α and HIF-2α, has been shown to inhibit low-glucose induced PCK2 expression in normoxia [71], on the other hand, hypoxia reduced PCK2 protein in lung cancer cells [90], and PCK2 was found to be inversely correlated with the hypoxia-induced glucose transporter GLUT1 in human lung cancers [90]. Recently, S-nitrosylation of PCK2 and the upstream enzyme PC (Figure 2) have been described to occur in different cancer cell lines in medium lacking glucose, in a manner dependent on nitric oxide production by the arginine-citrulline cycle [87]. This post-translational modification enhanced serine, glycine and thereby purine formation from glutamine in glucose-starved cancer cells [87]. Thus, oncogenic and nutrient-sensing pathways converge on the regulation of PCK1 and PCK2 in cancer cells, which ensures their flexible expression according to the nutritional state and metabolic demands.

In summary, gluconeogenesis enzymes PCK1 and PCK2 are expressed in cancers arising from diverse tissues, contrary to previous assumptions. Given their central position in carbon metabolism, regulating the OAA-PEP-pyruvate node, PCK1 or PCK2 might play an important role in modulating metabolism and cell fate decisions in many different cancer types, especially in a glucose-poor microenvironment. Since gluconeogenesis plays an important role in whole-body glucose homeostasis under starvation, the question arises whether inhibiting PEPCK would have any systemic side effects. While whole-body homozygous PCK1 knockout mice die early after birth [91], homozygous PCK2 knockout mice develop normally, but display impaired insulin secretion and oral glucose tolerance test [92]. A dual PCK1 and PCK2 inhibitor that was efficient in reducing subcutaneous colon cancer growth in vivo was recently found to be well-tolerated without inducing weight loss, albeit slightly reducing basal glucose levels in starved mice [93].

## 9. Fatty Acids as Fuels and Constituents of Structural Lipids

Oncogenes stimulate the de novo biosynthesis of fatty acids, the main constituents of neutral lipids and membrane phospholipids by enhancing the generation of fatty acid precursors like acetyl-CoA and malonyl-CoA and by upregulation of fatty acid synthase (FASN) and other enzymes [94]. Inhibition of FASN has been shown to perturb membrane composition and lipid raft assembly, reduce palmitoylation of signaling proteins and affect the activity of the phosphoinositide 3-kinase (PI3K)-AKT oncogenic pathway [95]. Thereby, inhibition of fatty acid synthesis induces tumor cell apoptosis and enhances cancer cell sensitivity towards chemotherapy [96]. However, besides synthesizing lipids de novo, cancer cells acquire lipids or fatty acids from the tumor microenvironment. Hypoxic cells have been shown to bypass de novo lipogenesis and fatty acid desaturation, a process mediated by the oxygen-dependent enzyme Δ9 stearoyl-CoA desaturase 1, by scavenging fatty acids [97]. Also, oncogenes like H-RAS lead to increased fatty acid uptake [97]. A lack of biosynthetic precursors for fatty acid synthesis, especially glucose or glutamine, has been proposed to stimulate fatty acid acquisition (reviewed in [98]). In fact, external addition of palmitate has been shown to be sufficient to restore decreased cell viability after inhibition of fatty acid de novo synthesis [99]. Fatty acids may be derived from circulating lipids by the action of lipoprotein lipase and monoacylglycerol lipase expressed by cancer cells [98]. Their expression in cancers correlates with tumor invasiveness (reviewed in [24]). However, cancer cells can also reprogram adipocytes and cancer-associated fibroblasts in their proximity to release fatty acids. Coculturing of adipocytes and breast cancer cells led to lipolysis in adipocytes and subsequent uptake of stable isotopically labeled fatty acids by breast cancer cells, which increased proliferation and migration [100]. Lipids released by stroma cells may also alter cell signaling besides contributing to metabolism. Lipid metabolites released from fibroblasts have been shown to enhance proliferation in prostate cancer cells and migration in colorectal cancer cells [101,102]. Recently, the lymph as a cancer microenvironment with a specific lipid profile came into focus. Oleic acid derived from the lymph has been shown to increase the metastatic potential of melanoma cells due to reduced oxidative stress and reduced ferroptosis [14]. 

Fatty acid uptake occurs via active transport; however, short-chain fatty acids can also enter passively through the cell membrane [98]. CD36, a translocase that mediates lipoprotein and fatty acid uptake, fatty acid transport proteins (FATPs) and fatty acid binding proteins (FABPs) all contribute to lipid/fatty acid uptake. CD36 expression has been shown to be correlated with poor survival in patients with lung squamous cell carcinoma, bladder cancer, luminal A breast cancer and ovarian cancer [16,103]. Expression of CD36 was enhanced by coculturing cancer cells with adipocytes or by a high fat diet and increased their metastatic potential [16,103]. Treatment with a CD36 antibody inhibited metastasis [16,103]. FABP4 deficiency substantially impaired metastatic ovarian cancer growth in mice and FABP was especially expressed at the interface between the adipocytes and the tumor [17]. Flexibility to activate fatty acid uptake instead of fatty acid biosynthesis was recently shown in MYC-induced liver tumors bearing a deletion of FASN [104]. The tumor cells imported fatty acids from the circulation and reducing fat in the diet synergized with FASN deletion and delayed tumor growth [104].

The role of hydrolysis of intracellular storage lipids in cancer is a matter of debate [105]. Recent studies showed that triglyceride degradation by hormone-sensitive lipase inhibits pancreatic cancer metastasis [106], while ABHD6, a monoacylglycerol lipase, promotes non-small cell lung cancer growth [107]. Triglyceride deposition in the form of lipid droplets appears to be enhanced in cancer cells in response to hypoxia [108] or in renal cell carcinoma in the context of mutation-driven HIF-1α activation [109]. Mechanistically, lipid droplets were shown to be a source of mono-unsaturated fatty acids for incorporation into phospholipids [109] and to protect tumor cells from reactive oxygen species [108]. The turnover of membrane lipids may be an additional important source of fatty acids and lipid precursors. Phospholipase C, for example, acts on phosphatidylinositol 4,5-bisphosphate to generate inositol 1,4,5-trisphosphate and diacylglycerol. The latter is a precursor for biosynthesis of different classes of phospholipids [110]. Hydrolysis of membrane phospholipids also yields free fatty acids. Thus, cancer cells acquire fatty acids from the tumor microenvironment or by hydrolysis of cellular lipids, and the fatty acids are used for energy production or de novo lipid biosynthesis (Figure 3). Fatty acids are no net gluconeogenic precursors in animals, yet the carbons from fatty acid derived acetyl-CoA are exchanged in the TCA cycle and may contribute to glucose or glycolysis intermediates via gluconeogenesis [111].

## 10. Autophagy and Macromolecule Degradation Provide Metabolic Intermediates in Glucose-Starved Cancer Cells

Cancer cells are not only avidly secreting proteins, including extracellular matrix proteins, into the microenvironment, they also take up proteins and other macromolecules. Certain cancer cell types, e.g., pancreatic cancer cells, are capable of importing macromolecules like albumin [112,113]; or they even digest larger complexes like cellular debris originating from cell corpses [114]. The uptake of macromolecules proceeds by engulfing extracellular material via micropinocytosis, or in a receptor-mediated manner, and is highly upregulated under nutrient deprivation (reviewed in [115]). Macropinocytosis is stimulated by oncogenic Ras [112] and also by Yap/Taz [116]. Nutrient deprivation additionally may activate autophagy, the process of self-digestion, which involves the degradation of organelles in autophagolysosomes [117]. Autophagy promotes cancer cell survival under nutrient shortage but also contributes to homeostasis and organelle quality control. Thus, it has been implicated as an adaptive strategy in nutrient-starved tumor cells, however, its overall contribution to cancer growth is highly context-dependent [117,118,119]. Catabolism of macromolecules, either derived from the extracellular space or from autophagy, yields amino acids or nucleotides that may be reused for anabolic purposes in so-called salvage pathways or oxidized for energy production (Figure 3). An indirect contribution of amino acids derived from protein degradation to gluconeogenesis has been observed in lung cancer cells [79]. In this study, yeast proteins were shown to be pinocytosed and hydrolyzed by cancer cells. The free amino acids, mainly alanine, accumulated in tumor cells and, in part, were shuttled towards gluconeogenesis via pyruvate and acetyl-CoA [79]. Thus, macromolecules may be a rich source for biosynthetic precursors in glucose-deprived cells facing a shortage of soluble nutrients. 

## 11. Metabolic Alterations in Immune Cells under Glucose Starvation

The metabolic microenvironment is shared by tumor cells, non-neoplastic stroma cells, like fibroblasts and endothelial cells, as well as by protumorigenic and antitumorigenic immune cells. This coexistence is characterized by cooperation on the one hand, and by competition for nutrients on the other hand. Recent studies in this rapidly growing field of research have been highlighted in excellent reviews [52,120]. The expression of PCK1 and partly also fructose-1,6-bisphosphatase 1, the downstream gluconeogenesis enzyme, has been found to be upregulated in monocytes cocultured with colon carcinoma cells [121]. This was associated with an increase in prostaglandin E2 synthesis and a net release of glucose by the monocytes [121]. This study suggests that gluconeogenesis might be activated in local macrophages/monocytes to promote glucose delivery to cancer cells. Interestingly, glucose limitation induced the release of cytokines and chemokines from different tumor cell lines, including interleukins IL-8, IL-6 and IL-2 [122]. The cytokines/chemokines in starved tumor cell supernatants, in turn, attracted innate immune cells, both in vitro and in vivo [122]. Highly proliferative activated T-cells have been shown to rely on glycolysis and lactate export [51,52]. Their proliferative capacity is compromised in conditions of low glucose and high lactate [51,52]. Recently, memory T cells were shown to upregulate PCK1 under glucose deprivation and to store carbons from amino acids as glycogen [123]. Together, these studies exemplify the necessity to consider both the metabolic reprogramming of cancer cells and tumor-infiltrating immune cells. Therapies modulating metabolic flexibility may also affect tumor-infiltrating immune cells in an undesired way.

## 12. Cell Signaling under Starvation—Orchestration of the Adaptive Response

Glucose and amino acid starvation elicit important homeostatic stress responses in cells. The integrated stress response pathway is triggered by the activation of the kinase GCN2 by different stimuli, including low amino acid or glucose levels, but also by oncogenes [124,125]. GCN2, which is activated by uncharged tRNAs [126], phosphorylates the eukaryotic translation initiation factor 2 (eIF2α) on serine 51, thereby inhibiting protein translation of most mRNAs, with the exception of stress-induced transcription factors, importantly, ATF4. ATF4 in turn leads to homeostatic responses by enhancing amino acid biosynthesis and uptake. The latter affects essential amino acids, like leucine, but also cysteine, a limiting factor for the biosynthesis of the antioxidant glutathione [126]. As mentioned, the integrated stress response and ATF4 are also critical activators of gluconeogenesis [70,127]. Mice with a homozygous mutation at the eIF2α phosphorylation site die soon after birth due to deficient gluconeogenesis and resulting hypoglycemia [127]. Additionally, ATF4 enhances lipogenesis [128]. GCN2 is not the only kinase activating ATF4. PERK, the endoplasmic reticulum stress-activated kinase, double-stranded RNA-dependent protein kinase (PKR) and heme-regulated eIF2a kinase (HRI) also activate ATF4 translation via eIF2α phosphorylation in response to cellular stresses [125]. The GCN2-eIF2α-ATF4 and PERK-eIF2α-ATF4 pathways can be cell-protective by initiating adaptive responses, however, they may also sensitize to cell death, among others, by ATF4-induced activation of CHOP. CHOP in turn leads to the expression of proapoptotic factors and TRAIL receptor 2 (TRAIL-R2/DR5) (reviewed in [124]). Silencing of ATF4 or GCN2, however, reduced proliferation and clonogenic growth in different cancer cell lines in media lacking certain nonessential amino acids or glucose and clearly reduced xenograft growth [129]. Interestingly, cell cycle arrest induced by a lack of amino acids was accompanied by reduced expression of asparagine synthase and restored by exogenous asparagine, but not by supplementation with other amino acids [129].

The cellular energy status is sensed by AMP activated protein kinase (AMPK), an important regulator of cell metabolism (reviewed in [119]). AMPK is a heterotrimeric complex formed from catalytic α-subunits and regulatory β- and γ-subunits. In response to decreasing ATP levels, the up-stream kinases LKB1 (encoded by STK11) and CaMKKβ (Ca^2+^/calmodulin-dependent protein kinase β) phosphorylate the α-subunit of AMPK on Thr172 and this, together with AMP bound to the γ-subunits, activates AMPK. To maintain energy homeostasis, AMPK shuts down energy consuming processes, for example it inhibits the anabolic fatty acid *de novo* biosynthesis pathway by phosphorylation of acetyl-CoA carboxylase while it enhances fatty acid oxidation. Importantly, AMPK activates autophagy, by direct phosphorylation of ULK1 [130]. In the liver, AMPK inhibits gluconeogenesis which is an anabolic, energy consuming process, via phosphorylation and nuclear exclusion of the CREB coactivator TORC2 (transducer of regulated CREB activity 2, also known as CRTC2) [131]. 

Mammalian target of rapamycin complex 1 (mTORC1) is another important signaling node influenced by glucose but also by amino acid levels. This complex consists of the core components mTOR, a serine/threonine protein kinase, Raptor (regulatory protein associated with mTOR) and mLST8 (mammalian lethal with Sec13 protein 8) [132,133]. High levels of glucose or high intracellular concentration of leucine activate mTORC1. mTORC1 controls the balance between anabolic and catabolic pathways in response to nutrient availability by regulating the biosynthesis of macromolecules while suppressing catabolic pathways such as autophagy (reviewed in [133]). The glycolytic intermediate dihydroxyacetone phosphate has just recently been identified as the key metabolite that is sensed to activate mTORC1 in glucose-replete conditions [134]. Downstream effectors of mTORC1 are kinases activating mRNA translation, importantly p70S6 kinase 1 (S6K1) and eIF4E binding protein (4EBP). Moreover, mTORC1 activates lipid biosynthesis by activating SREBP and promotes glycolysis by activating the translation of HIF-1α [133]. Active mTORC1 disrupts the interaction of AMPK and ULK1 thereby inhibiting autophagy [130]. Thus, the relative activity of mTORC1 and AMPK largely determines the extent of autophagy induction [133]. Nutrient availability is an important determinant of mTORC1 activity, however, growth-factor induced signaling pathways also play an important role. They converge on a key negative regulator of mTORC1 signaling, the tuberous sclerosis complex (TSC) [133]. In contrast to the well-known environment-dependent activation of mTORC1, mTORC2 is activated by G-protein coupled receptors via the small GTPase KRas4b [135]. mTORC2 promotes the activation of AKT and thus modulates metabolism and migration [133]. Both mTOR complexes play an important role in cancer growth [136]. Inhibitors of mTORC1 have been approved for the treatment of advanced renal cell carcinoma and other cancers, yet response rates have been rather low in some studies [136]. Studies with mTORC1 inhibitors involving biomarker-based patient selection and or combinations with other drugs are on the way [136]. 

Despite the large body of knowledge on the mechanisms of mTORC1 activation in cancer cells, little is known about the regulatory function of mTORC1 to mediate metabolic plasticity. In fact, tumorigenic mutations and the phenotypic program activated by these mutations (e.g., proliferation, invasion) may render cancer cells specifically dependent on certain metabolic pathways, thereby limiting their flexibility. Accordingly, hyperactivation of mTORC1 by growth factor pathways has been shown to increase the sensitivity towards glycolysis inhibitor 2-deoxyglucose [137]. 

## 13. Targeting Metabolic Flexibility as Anticancer Strategy

Antimetabolites targeting nucleoside biosynthetic pathways have been in clinical use for cancer chemotherapy for almost 70 years [138]. The understanding of the enzymes and pathways involved led to continuous improvement of antimetabolite agents and therapeutic regimes. The development of agents targeting other central metabolic liabilities in cancer, like glycolysis or glutamine addiction, however, is still in its infancy. To selectively block biomass and/or energy production in cancer cells, thereby promoting cell death, is one of the main goals of antimetabolic therapies (Figure 4). The enhanced activation of metabolic activities, an important hallmark of cancer cells, may be the basis for a selective antitumorigenic effect of such treatments, sparing normal tissues. As outlined above, metabolic flexibility allows the continued synthesis of crucial biosynthetic precursors, like glycolytic or TCA cycle intermediates from alternative sources when the prime precursors are missing. Thus, metabolic flexibility may limit the efficacy of antimetabolic therapies. Certain treatments targeting metabolism that are effective in vitro might be ineffective in vivo due to the utilization of different sets of nutrients [25]. The simultaneous inhibition of different key metabolic pathways activated in cancer cells in an individual tumor might be of advantage in order to account for the flexible switching between different metabolic phenotypes. A model depicting such an approach is shown in Figure 4. In line with such a concept, simultaneous inhibition of respiratory complex I and glycolysis inhibited melanoma progression [139]. In another study, synergistic inhibition of both glutaminases, the initial enzymes in glutamine utilization, and amidotransferases, which mediate alternative reactions, was required to block glutamine catabolism and growth of mouse and human tumors in vivo [104]. Importantly, metabolic flexibility can potentially be targeted by directly inhibiting cellular stress response pathways. Inhibition of GCN2, for example, reduced the expression of asparagine synthase in leukemia cells and thereby greatly enhanced their sensitivity towards asparagine depletion by asparaginase treatment, both in vitro and in vivo [140]. Interestingly, reprogramming of macrophages by cancer cells has been shown to play a role in adaptation to metabolic stress and interfering with such response-induced synergistic effects. For example, cancer cells deficient in complex I of the respiratory chain, as a mimic of respiratory chain inhibitor treatment, attracted protumorigenic M2 macrophages and simultaneous inhibition of protumorigenic macrophages, and complex I showed synergistic antitumor responses [141]. These examples show that eliciting metabolic stress in cancer cells and simultaneously targeting adaptive responses might prove effective in the treatment of certain cancers.

As a potential drawback to antimetabolic therapies, extreme stress may lead to selection pressure favoring the emergence of more aggressive clones. The capacity of murine melanoma and sarcoma cells to form lung metastases was clearly enhanced after exposure to glucose starvation for 48 h followed by a recovery period [142]. The underlying mechanisms have not been studied, however, the induction of stress-related proteins has been proposed by the authors to enhance the cells’ metastatic potential. In another study, low-glucose treatment of mismatch-repair deficient cancer cells selected for clones with higher expression levels of the glucose transporter GLUT1, which were retained also in high-glucose conditions. Partly, these cells showed a novel mutation of KRAS, which favored a high level of expression of GLUT1, thereby facilitating glucose uptake [26]. Based on these findings it has been suggested that a selection of pre-existing subclones by low-glucose conditions may contribute to metabolic variability in tumors [8]. In both instances, genetic adaptation or phenotypic adaptation, the activity of metabolic rescue pathways is critical for cell survival and proliferation during metabolic stress.

Of note, genetic alterations may also reduce the adaptability of certain cancers. Sequencing of human tumors revealed that deletions of tumor suppressors may involve the co-deletion of genes in their vicinity encoding for isoforms of metabolic enzymes, thus limiting metabolic flexibility in a specific manner. This dependence on certain enzyme isoforms may create synthetic lethality for inhibition of the “remaining” isoform of this enzyme. In pancreatic cancer, for example, malic enzyme 2 (ME2) is frequently co-deleted along with the SMAD4 tumor suppressor, rendering tumor cells dependent on the only remaining mitochondrial malic enzyme isoform, ME3 [143]. Thus, to target metabolic dependencies created by (co-)deletions of specific enzymes, which per se limits metabolic flexibility in a cancer-specific manner, is a promising concept. 

## 14. Summary and Outlook

In solid tumors, cancer cells are forced to adapt to changing nutritional and oxygen conditions due to regional differences in blood perfusion, tissue architecture and other factors, as outlined above. The flexible use of nutrients by cancer cells, which partly proceeds via nonclassical pathways, e.g., the gluconeogenesis pathway, not only promotes their dissemination and tumor evolution, it may also greatly affect the efficacy of antimetabolic therapies. Blocking one metabolic pathway may elicit adaptive responses in cancer cells favoring alternative pathways, ultimately leading to therapy resistance. Thus, simultaneous targeting of multiple metabolic pathways might show synergistic responses, as reported in several studies. Moreover, inhibiting signaling pathways enhancing adaptation or blocking central metabolic routes that are directly involved in mediating flexibility, like the gluconeogenesis pathway, might prove effective in future therapeutic approaches. A better understanding of metabolic flexibility in cancer cells may help identify biomarkers guiding possible antimetabolic therapies and to uncover mechanisms of resistance towards such treatments.

## Figures and Tables

**Figure 1 ijms-22-01476-f001:**
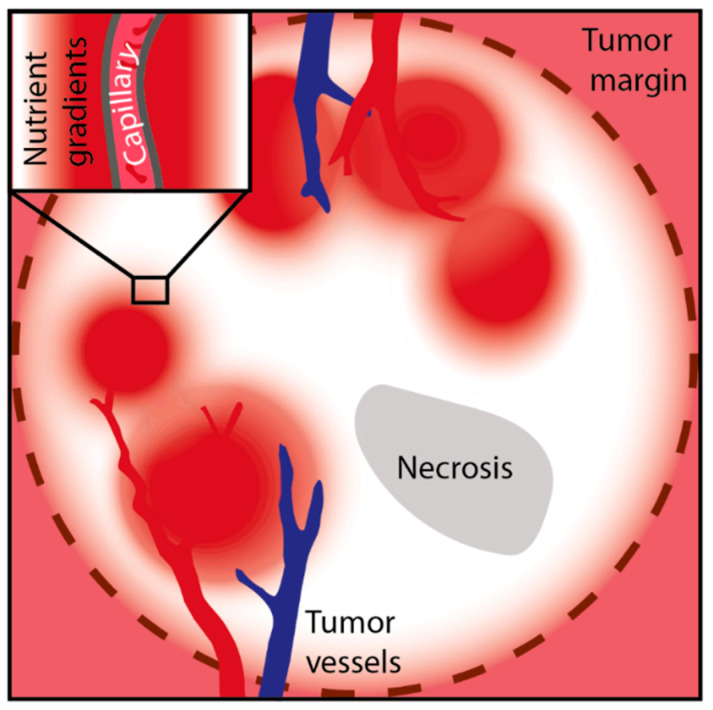
Heterogenous perfusion and nutrient supply in solid cancers. Typical pattern observed in scans of perfusion magnetic resonance imaging, showing regions of high (red) and low blood perfusion (white) scattered around solid cancers. The vascular network, comprising tumor arteries (red), veins (blue) and the connecting microvessels, is frequently chaotic, resulting in fluctuating, insufficient blood flow. On a microscopic scale, steep gradients for nutrients and oxygen with increasing distance from the capillaries occur (detail).

**Figure 2 ijms-22-01476-f002:**
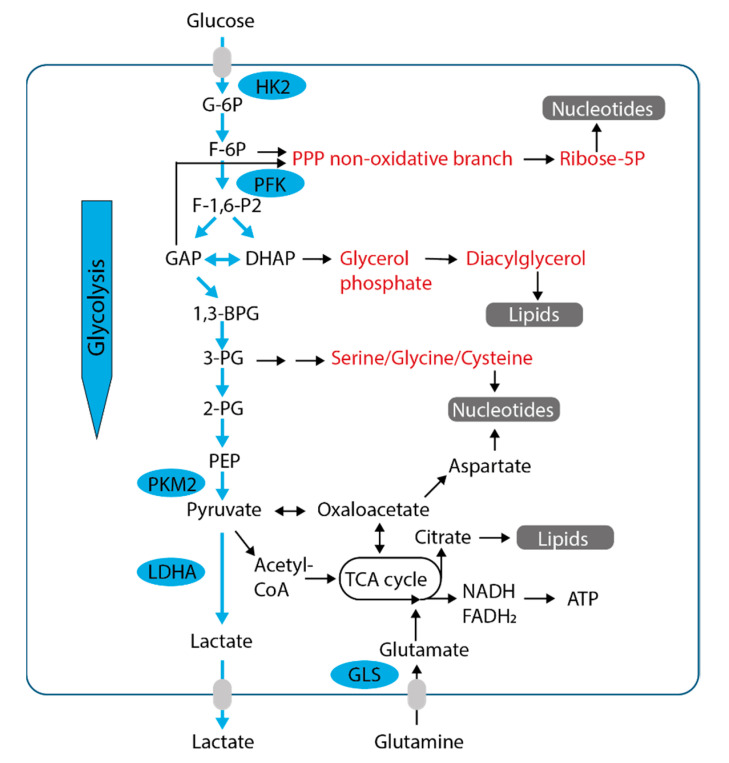
Biosynthetic pathways branching from glycolysis in glucose-replete conditions. Glucose is the main precursor for glycolysis (blue arrows). Glycolytic enzymes are upregulated in many cancer types, leading to enhanced generation of glycolytic intermediates. These serve as precursors for numerous metabolic pathways (marked in red) necessary for the biosynthesis of cellular biomass (highlighted in grey). Excess glycolytic intermediates are converted to pyruvate and finally lactate, which is released. The conversion of pyruvate to acetyl-CoA, leading to full oxidation in mitochondria, is relatively decreased. Additionally, amino acid catabolism, primarily glutaminolysis, is required to meet the cancer cell’s demand for nitrogen and biosynthetic precursors. PPP, pentose phosphate pathway; HK2, hexokinase 2; PFK, phosphofructokinase; PKM2, pyruvate kinase M2; LDHA, lactate dehydrogenase A; GLS, glutaminase; TCA cycle, tricarboxylic acid cycle.

**Figure 3 ijms-22-01476-f003:**
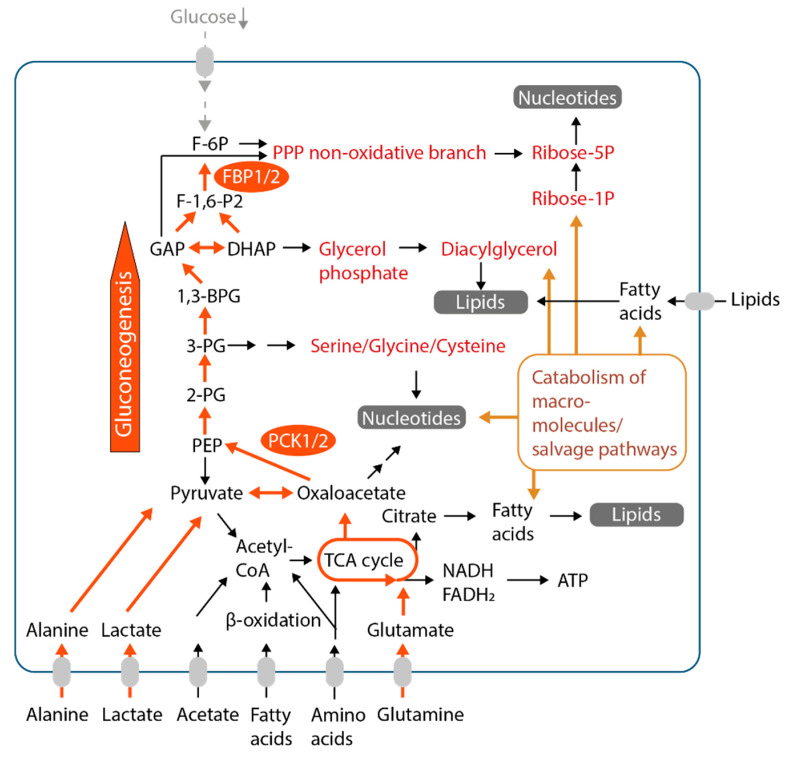
Gluconeogenesis and other rescue pathways for the generation of metabolic intermediates in low-glucose conditions. Precursors for biomass biosynthesis are produced from multiple alternative nutrients, like lactate or amino acids or from degradation of macromolecules (via autophagy or macropinocytosis, brown arrows/box) during a lack of glucose. Initial steps of the gluconeogenesis pathway (orange arrows), mediated by PCK1 or PCK2, are key for the generation of glycolytic intermediates under low-glucose conditions. These may serve as precursors for different metabolic pathways (marked in red) necessary for the biosynthesis of cellular biomass (highlighted in grey) also in the absence of glucose.Furthermore, the TCA cycle and enzymes of pyruvate carboxylation and degradation play an important role in the degradation or interconversion of alternative, noncarbohydrate precursors/nutrients according to the cell’s demands. TCA cycle, tricarboxylic acid cycle, Krebs cycle; PCK1/2, phosphoenolpyruvate carboxykinase cytoplasmic isoform (PCK1) or mitochondrial isoform (PCK2); FBP1/2, fructose-1,6-bisphosphatase 1; PPP, pentose phosphate pathway.

**Figure 4 ijms-22-01476-f004:**
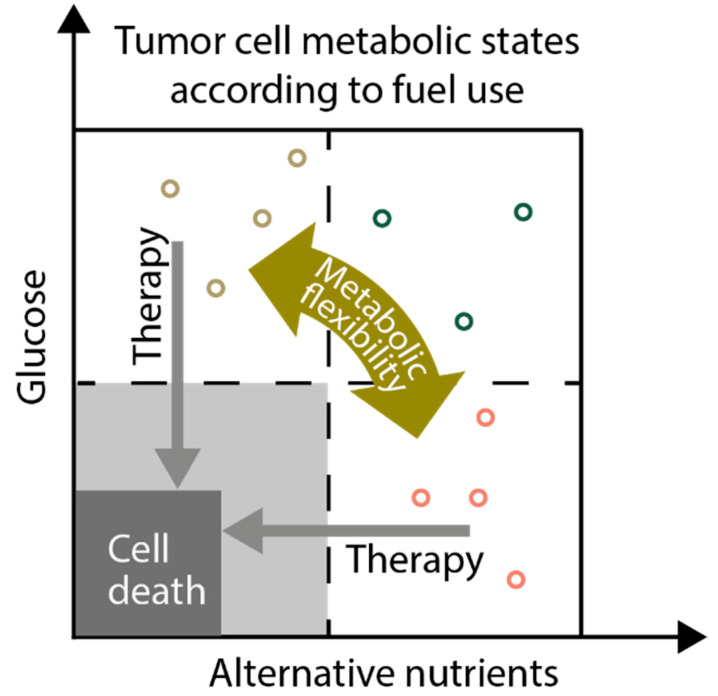
Targeting aberrant cancer metabolism and metabolic flexibility as an anticancer strategy. As an example of cancer cell metabolic flexibility, cancer cells (represented as circles) may switch from a state of high glucose consumption/glycolysis (brown circles) to the use of alternative nutrients (e.g., lactate; symbolized by red circles) according to the availability in the tumor microenvironment or in response to other factors. Green circles in the right upper quadrant represent cells consuming abundant glucose and “alternative” nutrients simultaneously. Treatments targeting metabolic pathways in cancer cells, like glycolysis, should take metabolic flexibility into account. Inhibition of multiple, compensatory metabolic pathways simultaneously might be necessary to reduce proliferation (light grey) or induce cell death (dark grey).

## Data Availability

Not applicable.

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
