# Peer review of "Flexibility and Adaptation of Cancer Cells in a Heterogenous Metabolic Microenvironment"

_ijms, 2021, doi:10.3390/ijms22031476_

Round 1
Reviewer 1 Report
Line 26 – stroma is referred only to connective tissue cell types– if the authors want to include all non-malignant populations in tumor mass (including immune cells), they should avoid the use of word stroma (the same commenta valid also in the rest of the review, for example in line 60 I suggest to change “non-neoplastic stroma cells” into: “non neoplastic populations of the tumor microenvironment”)
Specific comments:
Line 38 – the authors should be more specific in what they mean by the fact that cancer cells asre similar to normal cells – which normal cells and in which context?
Line 41 – it is not clear to what is the word “these” referring to
Line 80-82 – I suggest not to separate the extrinsic from intrinsic factors - cancer progression follows rules of darwinian selection, so one may say that nutrient availability is a selective pressure that, as the authors state, determines metabolism of a cancer cell, but most likely also the selection of oncogene/tumor suppressor mutations that promote the biochemical reactions leading to required metabolism
Figure 1 – I suggest using 3 colors: red for high perfusion; white for low perfusion and maybe dark gray for necrosis – to distinguish the 3 conditions so they don’t have to be written in the figure more than once-ot the text can be completely removed as the color code can be explained in the caption. Also, the gradient should be represented better in the detail. In general, the resolution of the figure should be improved
Line 88 – the necrosis is written necroses + it is redundant to say twice that necrosis is mainly occuring in the tumor center
Line 151 – precursors instead of precursor
Line 152 – indicate the chapter number
Figure 2- There is text indicated with different colors without explaining in the caption what each color is being referred to. Also, in the line 160-it is not clear which precursors the authors refer to?
It is a good idea to present a figure that shows pathways activated upon glucose shortage. However, to a naïve reader, the way the Figure 2 is organized it merely seems a set of reactions taken from a biochemistry textbook. It is the most important figure of the review, it should be done more clearly. For example, glycolysis is shown, as if it were a part of alternative pathways, when glycolysis is actually the canonical pathway the glucose would be used through. The authors should provide a more explicit representation of the alternative pathways, even if it means reducing the number of shown reactions/metabolites in each biochemical process, and even not representing the biochemical reactions at all, but rather just connect metabolites with lines representing the alternative pathways. I would actually suggest to divide it in 2 separate figures as I explain below in general comments.
Line 178 – “is” is missing between It and “taken up”
Chapter 6 is entitled “glutamine and other aminoacids”, but the chapter deals with glutamine only. I understand that the authors only wish to give an idea of what aminoacid role in anaplerosis can be. Still, either the title of the chapter should be changed or more than one example of aminoacids as anabolites should be given.
Chapter 7 – the extent to which the authors write about lactate is much broader than what the title indicates – I understand there is so much literature on lactate and it is not trivial to fit everything in only one chapter, but if the authors decided to focus on lactate as a fuel than several topics they tackle in this chapter are out of scope – this chapter should be shortened and more focused on the role of lactate as a fuel (there are myriad of reviews dealing with the rest of lactate’s role in supporting cancer)
Line 252 – I suggest a separate figure on gluconeogenesis, not only because in figure 2 it is not very clearly represented, but also since it is the main expertise and novelty of the review
Chapter 9 – at least a comment on the role of lipid droplets in the context of cancer should be mentioned in this chapter
Line 397 – since, as I understood, the gluconeogenesis part is the novelty of this review, it would be worth explaining a bit better what this cited paper is dealing with and possibly giving an author-based comment – an opinion on what this could mean in a more general context
Figure 3 – Again, it is not clear what differently colored circles refer to, in general the authors should explain in the caption what is the graph representing cancer growth? Therapy? Nutrient usage? It is not intuitive or sufficiently clear, at least to me
General comments:
It is a comprehensive review, which is a difficult task since there is a lot of literature on the topics the authors tackle, and they mostly do a good job in summarizing each topic. I would suggest a rearrangement of the chapters, dividing it in 3 parts, and dividing also figure 2 in two separate figures:
- Introductory part that would involve current chapters: 1,2,3,4, 5 and 11 – explaining the framework within which the alternative fuel metabolites will come in the picture – here a figure could be made showing BASIC processes of cancer metabolism
- Alternative fuel part with chapters: 6, 7, 8, 9 and 10 - here a figure dealing ONLY with alternative pathways to overcome nutrient deprivation could be prepared
- Therapy prospects as chapter 12
The figures 2 and 3 require major revision.
The authors should try to comment more on the data they present. Often, the chapters are concluded with a sentence saying “it is still not known a lot about…” – which is true, but it would be useful for the reader if the authors anticipated hypotheses about what is the possible implication of metabolic plasticity in the context they are writing about.
In my opinion, a chapter is missing that deals with how metabolic microenvironment influences the composition of non-malignant cell populations within a tumor. There is a lot of literature emerging on this topic and should be acknowledged in this review. Especially since the cancer cell cooperation or even parasiting on neighbouring cells is a well-known phenomenon and can be considered as one of the alternative sources of fuel in glucose deprived conditions. The lines 236-239 could be transferred to such a chapter. Also, when dealing with simultaneous targeting of various metabolic adaptations in line 467, the authors should cite 30796225, where an improvement of treatment was obtained by simultaneous targeting of metabolism in a cancer and non-malignant cell populations.
Reviewer 2 Report
The review manuscript by Gabriele Grasmann et al touches upon important topics that can be of interest for researchers working in the field of metabolism, since I consider the metabolic flexibility of cancer cells a hot topic. The review is detailed, and the references used are appropriate. The review is well written, enjoyable, and the language is acceptable. Although the manuscript is almost ready for publication, below I list a few suggestions to improve the text.
1 – The authors have used in the present manuscript and previous works the term “abbreviated gluconeogenesis” which can be controversial, and which I consider not accurate. Gluconeogenesis is the de novo synthesis of glucose. This reviewer considers abbreviated a “consummated” (glucose producing) pathway that uses fewer steps. The pathway presented in cancer cells is “unconsummated”, thus it should not be referred to as abbreviated, but incomplete/partial, or not be re-termed. The authors explain the term, but I still consider it not accurate. For instance, (as properly explained there) glyceroneogenesis is not an abbreviated gluconeogenesis, it’s a consummated synthesis of glycerol, that shares a part of the pathway with the canonical gluconeogenic route.
2 – The authors did a good job detailing the alternative nutrients that cancer cells can use in diverse metabolic scenarios, however I think that acetate is a very relevant nutrient for cancer cells, and should be included.
3 – “Cell signalling under starvation -the orchestration of the adaptive response” paragraph is mainly focused in MTOR pathway. It would be interesting to include other nutrient-sensing branches such as the UPR and AAR responses (PERK/GCN2 – ATF4).
see: EMBO J. 2010 Jun 16; 29(12): 2082–2096.
4 – “However, it must be taken into account that inducing extreme stress on cancer cells may favor the selection for more aggressive clones [7]. “
The authors cite here a review to support their statement. However, if the same statement in that work is only supported by 1 study, the authors should cite that work, and or find and cite other supporting evidences.
5 – “Although glucose is an important nutrient for cancer cells, the poor vascular supply and high rate of consumption lead to a poor supply of this sugar in certain parts of solid cancers” could benefit from a reference, maybe reuse Refs 1, 2 or 42.
6 – Reference 124 can be re-used in section 9: “However, besides synthesizing lipids de novo, cancer cells acquire lipids or fatty acids from the tumor microenvironment” (dietary lipids support tumour growth in the absence of fasn).
7 - Authors state the canonical view that “The utilization of fatty acids for gluconeogenesis, however, is not possible in mammalian tissues, since acetyl-CoA is not a gluconeogenic precursor”. The sentence, although technically correct, should be referenced.
(However, although fatty acids are not a NET gluconeogenic precursor, the acetyl-CoA from short-chain fatty acids such as acetate can contribute to the glucose skeleton).
Round 2
Reviewer 1 Report
I appreciate the authors acknowledging most of the suggestions.
The only note I have is to remove the story and citation regarding neutrophils from lines 518-523, if the paper is not referring to tumor associated neutrophils.
Author Response
We highly appreciate this suggestion and deleted the respective reference on neutrophils.